# Separation of Microalgae by a Dynamic Bed of Magnetite-Containing Gel in the Application of a Magnetic Field

Takehiro Washino [1], Mikihide Demura [2], Shintaro Morisada [1], Keisuke Ohto [1] and Hidetaka Kawakita [1,*]

1 Department of Chemistry and Applied·Chemistry, Faculty of Science and Engineering, Saga University, Saga 840-8502, Japan; 20705030@edu.cc.saga-u.ac.jp (T.W.); morisada@cc.saga-u.ac.jp (S.M.); ohtok@cc.saga-u.ac.jp (K.O.)
2 Department of Agriculture, Saga University, Saga 840-8502, Japan; st8148@cc.saga-u.ac.jp
* Correspondence: kawakita@cc.saga-u.ac.jp

**Abstract:** Microalgae are now known as potential microorganisms in the production of chemicals, fuel, and food. Since microalgae live in the sea and the river, they need to be harvested and separated and cultured for further usage. In this study, to separate microalgae, a bed of magnetite-containing gel (Mag gel, 190 μm) was packed in the column by the application of a magnetic field for the separative elution of injected microalgae (including mainly four species), cultured at Saga University in Japan. The applied magnetic field was set at a constant and dynamic-convex manner. At a constant magnetic field of 0.4–1.1 T, the elution percentage of the microalgae at less than 5 μm was 30–50%. At 1.1 T, the larger-sized microalgae were eluted at a percentage of 20%, resulting in the structural change of the bed by the applied magnetic field. In a convex-like change of the magnetic field at 1.1 T $\rightleftarrows$ 0.4 T, the smaller-sized microalgae were selectively eluted, whereas at 1.1 T $\rightleftarrows$ 0.8 T, the larger-sized microalgae were eluted. Dynamic convex-like changes by the magnetic field selectively eluted the microalgae, leading to the separation and the extraction of potential microalgae.

**Keywords:** magnetite-containing gel; microalgae; separation; dynamic





## 1. Introduction

Several microalgae exist in large bodies of water and rivers. Microalgae have been studied to produce biopharmaceuticals, biofuel, and food [1–3]. The separation of certain microalgae and their subsequent culturation in a pure system allows us to produce chemicals at higher efficiency. However, microalgae are currently recovered by hand-use plankton nets and micropipettes [4]. More high-performance separation techniques of microalgae are required to realize their potential.

Thus far, for the separation of microorganisms, chromatographical columns packed with beads [5,6] and capillary electrophoresis [7,8] have been used. Though the chromatographical technique separates microalgae based on surface characteristics and sizes as well as morphology, separation mechanisms such as size effect and adsorption site in columns are homogeneous. Capillary electrophoresis separated the charged microorganism based on their mobility under the flow of mobile phases and their applied voltages. Since some operational conditions, such as flow rate, the gradient of concentration of salts, and species of beds, are key factors for separation in chromatography, the bed structure is not changed. Dynamic change of the bed structure separates the injected microorganisms with various modes of the bed to elute each microorganism based on the morphology, size, and surface characteristics, thus leading to sophisticated separation equipment.

Magnetite-containing materials in columns are able to change the macroscopic structure with an application of a magnetic field. Magnetite particles thus far have been used for harvesting microalgae [9] and purification of protein [10,11], polysaccharides [12], and cells [13]. A functional group was introduced to the surface of the magnetite particle. The obtained particle was added to the solution, and after the adsorption, the magnetic particle

was recovered by applying a magnetic field. This technique was mainly used in batch mode. Levenspiel studied the change of magnetite in a closed system in a magnetic field in a gas flow [14,15]. He summarized the relationship between the magnetic field and the Reynolds number of flow as a phase diagram. Hristov also considered the behavior of magnetite by fluid in columns, and their effects on magnetic strength, the size of magnetite, and fluid dynamics [16,17]. Kawakita's group separated starch granules and silica particles by the magnetite-assembled membrane with the on-off stimulus of the magnetic field in that the macroscopic pore-like structure of the magnetite-composed membrane dynamically intercepted and released the particles [18,19].

In this study, magnetite-containing gel (Mag gel) was packed in a column by applying a magnetic field for microalgae separation (Figure 1). Packed Mag gel forms the pore among the gels in the column. Flowed microalgae injected on the top of the bed of the Mag gel were separated by the formed pore-like gaps based on the size and morphology as well as the interaction. Compacted Mag gels have various gaps in the direction of flow due to the distributed solid pressure in height, having the pore distribution along with the column height. The distributed gaps of gel intercepted the various microalgae, as already studied by our group [20]. The applied magnetic field dynamically altered the gaps' structure of the gel bed in the column's flow direction and radius direction to elute the intercepted microalgae. Namely, the gaps among the gels were statically and dynamically altered by applying the magnetic field to elute the intercepted microalgae, which were selectively dependent on the size and morphology. The technique proposed thus far for the separation of microalgae could be improved for the potential application of producing chemicals by newly discovered microalgae.

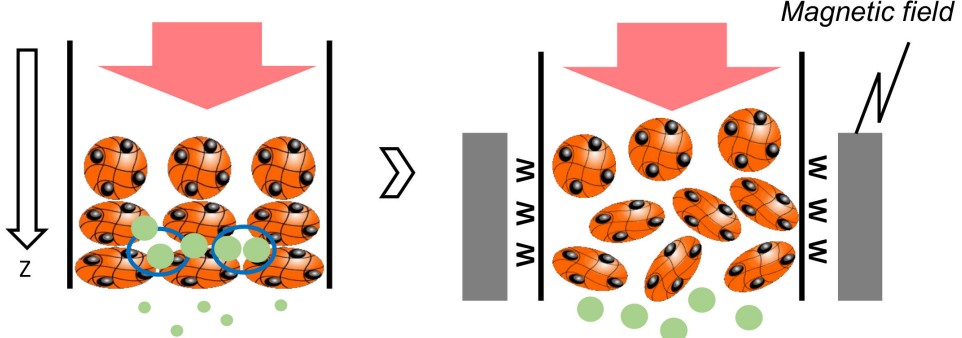

**Figure 1.** Dynamic Mag-gel bed by different magnetic fields for the separation of microalgae.

## 2. Materials and Methods

### 2.1. Materials

*N*,*N*-dimethylacrylamide (049-19185, Lot: KCE2867), *N*,*N*'-methylene bisacrylamide (134-02352, Lot: KCJ7454), and ammonium persulfate (018-03282, Lot: HWJ7442) were purchased from FUJIFILM Wako Pure Chemical Corporation, Tokyo, Japan. Ethylene diamine tetraacetic acid (09-1320, Lot: 0544) was obtained from Katayama Chemical Industries, Japan. Span80 (S0060, Lot: Z3MZIDD) and Tween 80 (T0546, Lot: QC3MHOE) was obtained from Tokyo Chemical Industry Co., Tokyo, Japan. Magnetite (15-1380-5) was obtained from Sigma-Aldrich, Inc., St Louis, MO, USA. The membrane filter (10000MWCO, polyethersulfone, 1165093) was purchased from Sartorius Stedim Biotech GmbH. Microalgae used in this study were cultured in the Department of Agriculture, Saga University, Japan (image of microalgae is shown in Figure 2). Other chemicals were of analytical grade or higher.

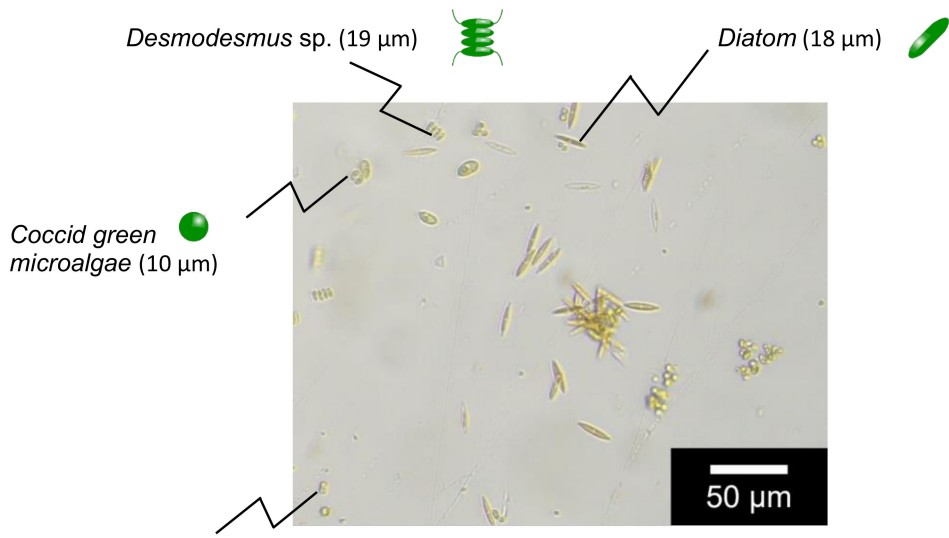

**Figure 2.** Image of target microalgae.

## 2.2. Preparation of Magnetite-Containing Gel

Magnetite-containing gel (Mag gel) was prepared by suspension polymerization in the presence of magnetite, as previous paper [20]. The amount of chemicals used is summarized in Table 1. *N,N*-dimethylacrylamide, *N,N*-methylene bisacrylamide (crosslinker), and ethylene diamine tetraacetic acid were dissolved in water. Span 80 and Tween 80 were dissolved in hexane in stirring at 120 rpm in a three-necked flask at 343 K in a bath (SB-350, EYELA Rikakikai Co., Ltd., Tokyo, Japan). Magnetite and ammonium persulfate were dissolved in water.

**Table 1.** Preparation condition of magnetite-containing gel.

|  | Chemical | Mass [g] | Mole [mmol] |
|---|---|---|---|
| | *N,N*-dimethylacrylamide | 9.4 | 94 |
| | *N,N'*-Methylene bisacrylamide | 0.161 | 1 |
| Water phase | Ethylenediamine tetraacetic acid | 0.148 | 0.53 |
| | Ammonium persulfate | 0.335 | 1.4 |
| | Distilled water | 25.2 | 138 |
| | Magnetite | 2.06 | 8.63 |
| | Span 80 | 9.28 | 21.5 |
| Organic phase | Tween 80 | 3.44 | 2.59 |
| | hexane | 100 | 116 |

In the water phase, monomer and magnetite-containing solutions were dropped in the oil phase, stirring at 120 rpm at 343 K for four hours. After the polymerization, the product was filtered in vacuo. The product was washed with hexane, ethanol, and water. The obtained product was weighed, and water was added to the product at a concentration of 40 wt%. The morphology of the gel was observed by optical microscopy (VH-SS, Keyence Corporation, Japan) to determine the size distribution by counting more than 200 numbers of the gel's size. The magnetite content in the obtained Mag gel was determined by dissolving Mag gel in 3 M $HNO_3$ solution at 343 K. The concentration of iron ions was determined by atomic absorption spectroscopy (AA-7000, Shimadzu Corporation, Kyoto, Japan). The state of the prepared Mag gel is shown in Supplementary Materials Figure S1. The mean size of obtained Mag gel was 190 µm. The existence of magnetite in Mag gel was checked with microscopy. The calculation method of the magnetite content is described in Supplementary Materials S2. The properties of the Mag gel are summarized in Table 2.

**Table 2.** Properties of Mag gel.

| Mean Size [μm] | 190 |
|---|---|
| size of manunetite used [nm] | 280 |
| number of magnetite in Mag gel * | $3.3 \times 10^6$ |
| Volume occupation percentage in Mag gel [%] * | 1.1% |

* calculation was mentioned in Supplementary Materials S2.

*2.3. Adsorption of Microalgae to Mag Gel*

Microalgae suspensions at several concentrations were mixed with Mag gel solution (0.68 mL, number of Mag gel: $6.9 \times 10^4$ number). The volume of the solution was from 1.7 mL to 2.7 mL. pH was set at 6.7, and adsorption was performed at room temperature. After the recovery of Mag gel with a magnet (0.10 T, 50 mm $\times$ 20 mm $\times$ 2 mm, Sangyo Supply Co., Ltd., Sendai, Japan), the remaining concentration of microalgae was determined by C-Chip (DHC-F01, ARBROWN Co., Ltd., Tokyo, Japan) with the observation of microscopy (BA81-6T-1080, Shimadzu RIKA Corp., Kyoto, Japan).

*2.4. Mag Gel Packed in Column by Application of Magnetic Field for Separation of Microalgae*

The Mag gel suspension (40 wt%) was inserted into a glass column (I.D. 1 mm) with the application of a magnetic field (TM103-15045-0560, Tesura Corporation, Japan) connected with a power supply (PAN35-5A, Kikusui Electronics Corp., Japan), thus packing the Mag gel in the column. The flow rate of water, the strength of the applied magnetic field, and the amount of Mag gel were changed to examine the packing stability of the bed from the following leakage percentage, as shown in Equation (1), of the Mag gel. The concentration of Mag gel in the elution was determined by C-Chip with microscopy.

$$\text{Leakage percentage of Mag gel [\%]} = 100 \, (\text{amount of leaked mag gel})/(\text{initial packed Mag gel}) \qquad (1)$$

The microalgae suspension, including mainly four species, was injected into the bed top of Mag gel (height: 15 cm). The numbers of each microalgae (under 5 μm microalgae, *Coccid* green microalgae, *Diatom*, *Desmodesmus* sp.) were $1.9 \times 10^4$, $3.7 \times 10^3$, $3.0 \times 10^3$, and $2.0 \times 10^3$, respectively. The applied magnetic field was set at a static constant strength up to 1.1 T and at a dynamic convex change of 1.1 T $\rightleftarrows$ 0.4 T, 1.1 T $\rightleftarrows$ 0.6 T, and 1.1 T $\rightleftarrows$ 0.8 T. The magnetic field in the dynamic convex manner was changed at each interval of 10 min. Water was permeated through the bed by a syringe pump (S-1235, Atom Medical Corp., Japan) at the various flow rates for 60 min. The scheme is illustrated below in Appendix A.

The form comparison with the bed of Mag gel and glass bead (diameter: 115 μm, 9 g) was packed in the column (I.D. 1 cm) at the height of 5 cm. The porosity of the bed was set at 0.65. The numbers of the injected microalgae (under 5 μm microalgae, *Coccid* green microalgae, *Diatom*, *Desmodesmus* sp.) were $1.4 \times 10^6$, $2.2 \times 10^5$, $2.8 \times 10^5$, $1.5 \times 10^5$, respectively. The flow of water was permeated by gravity, and the flow rate measured was 1.4 mL/min. The elution percentage of each type of microalgae was determined by Equation (2):

$$\text{Elution percentage [\%]} = 100 \, (\text{amount of eluted microalgae})/(\text{injected amount of each microalga}) \qquad (2)$$

**3. Results and Discussion**

*3.1. Leakage of Mag Gel from Column in Application of Magnetic Field*

Magnetite-containing gel (Mag gel) was packed into the column by applying the magnetic field for the separation of microalgae. To separate the microalgae in the flow through the column, Mag gel is needed to maintain the packed structure in the column for further separation (Supplementary Materials Figure S3). The leakage of Mag gel from the column is dependent on the strength of the applied magnetic field, flow rate, and amount of packed Mag gel. The relationship of leakage percentages with various above-mentioned conditions is shown in Figure 3. By increasing the water flow rate, the leakage percentage increased because of the increment of pressure and shear stress. The percentage decreased

by increasing the magnetic field, and the Mag gel could be packed in a column up to 1.1 T. A larger amount of Mag gel lessens the leakage percentage due to a lower gradient of fluid pressure. The experimental condition (e.g., above 0.4 T of magnetic field) without leakage of the Mag gel from the column was used in the following experimental setup of separating microalgae.

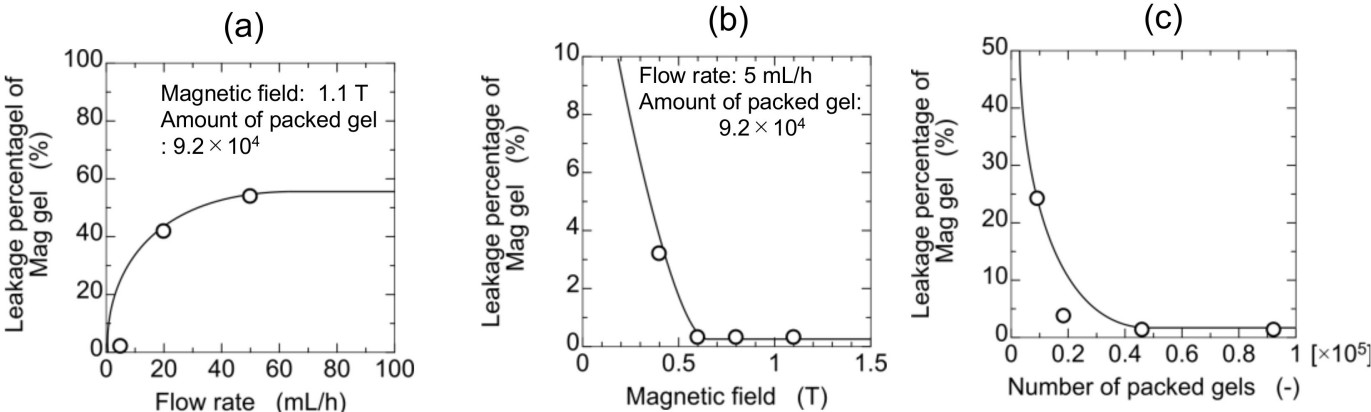

**Figure 3.** Elution of Mag gel from a gel-packed column in a permeation of water. (**a**) Effect of flow rate, (**b**) effect of the magnetic field, and (**c**) effect of a number of packed gels.

### 3.2. Adsorption of Microalgae to Mag Gel

Microalgae could be adsorbed into Mag gel in permeating the microalgae suspension to the bed of Mag gel. The mag gel was added to the microalgae suspension to determine the adsorption behavior. The relationship of the amount of microalgae adsorbed to the Mag gel with the concentration of microalgae at equilibrium is shown in Figure 4.

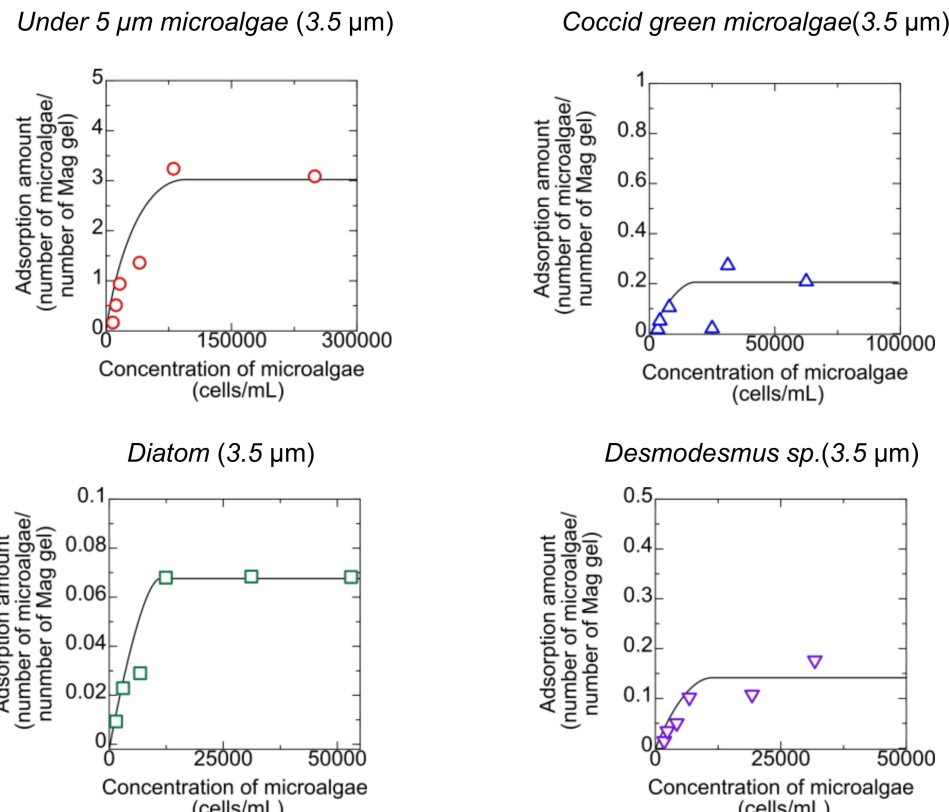

**Figure 4.** Adsorption of microalgae to the Mag gel.

The microalgae less than 5 μm adsorbed to the Mag gel at a maximum of three numbers per one Mag gel, meaning that the smaller microalgae adsorbed to Mag gel, whereas the other microalgae were adsorbed to one Mag gel less than 0.2 number of one Mag gel. This indicated that in the permeating suspension of microalgae to the bed of Mag gel, the large-sized microalgae would be intercepted by the size and shape effect and not by interaction. The gel has a three-dimensional structure, and the matrix of the Mag gel has a pore size of dozens of nanometers [21]. The smaller-sized microalgae did not move to the inner part of the Mag gel, resulting in its adsorption on the surface. Microalgae have proteins and polysaccharides on the surface [22,23], thus adsorbing the amide group on the surface of the Mag gel.

### 3.3. Separation of Microalgae in Static Mag Gel in Application of Magnetic Field

The mag gel was packed in a column at the constant magnetic field ranging from 0.4 to 1. 1 T. After the injection of the microalgae suspension, the water was permeated. During the permeation, no elution of the Mag gel was checked. Time-course curves of the elution of each type of microalgae are shown in Figure 5. The *Coccid* green microalgae and *Desmodesmus* sp. were not eluted, whereas the microalgae at less than 5 μm and *Diatom* were eluted. In the case of microalgae at 5 μm at 1.1 T, the elution percentage reached 35% and decreased with time. At a lower magnetic field, the elution percentage of the microalgae gradually decreased. Because of the linear shape of Diatom, the elution percentage was 15%.

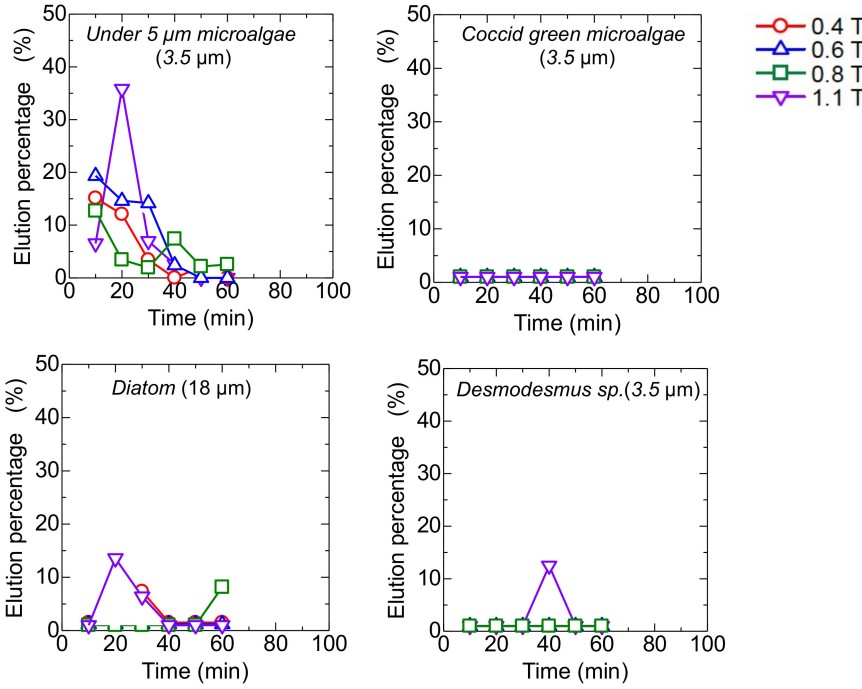

**Figure 5.** Time-course curves of eluted each type of microalgae from a bed of Mag gel in applying a static magnetic field.

The elution percentage up to 60 min was summarized in Figure 6. The percentage of microalgae at less than 5 μm was higher at each magnetic field strength. At 0.8 T, the *Diatom* eluted at 18%, and at 1.1 T, the *Desmodesmus* sp. also eluted. An illustrated image of the column at lower and higher magnetic fields is shown in Figure 7. At 0.4 and 0.6 T applied, the Mag gel was homogeneously packed in a column, the smaller-sized microalgae going through the gaps of Mag gel. At a higher magnetic field of 0.8 and 1.1 T, the Mag gel was heterogeneously packed to the radius direction of the column, resulting in the formation of larger gaps of Mag gel to elute the *Diatom* and *Desmodesmus* sp.

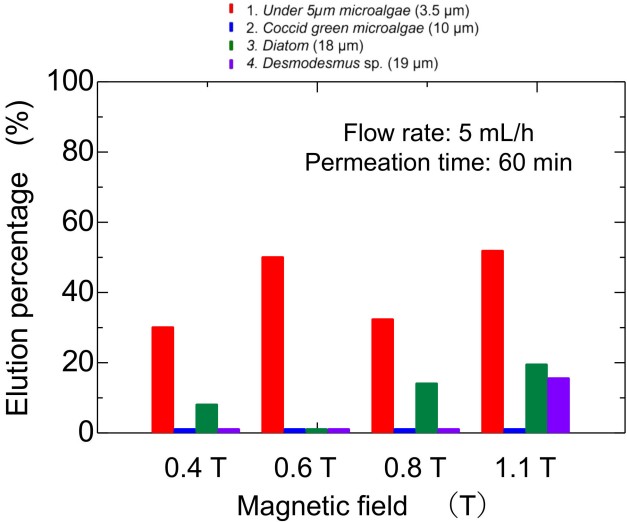

**Figure 6.** Elution percentage of microalgae at various static magnetic fields.

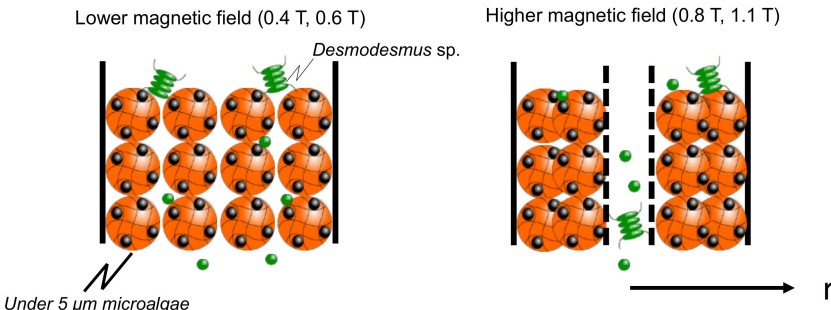

**Figure 7.** Illustrated image of elution of microalgae at lower and higher magnetic fields.

Glass beads (size: 115 μm) were used as a bed composition in a column to compare the elution behavior of the bed of Mag gel. Glass beads were homogeneously packed, and the gaps formed would be uniform. The time-course curve of the elution percentage and summed elution percentage are shown in Figure 8a,b, respectively. The smaller-sized microalgae were gradually eluted, and its total elution percentage was higher. The elution percentage of the *Coccid* green microalgae and *Diatom* were 18% and 16%, respectively, demonstrating that no selectivity was obtained. As shown in Figure 6, at 0.8 T, the elution percentage of *Diatom* was higher, meaning that the deformation of Mag gel by applied pressure and radial distributed by the magnetic field enhances the separation performance.

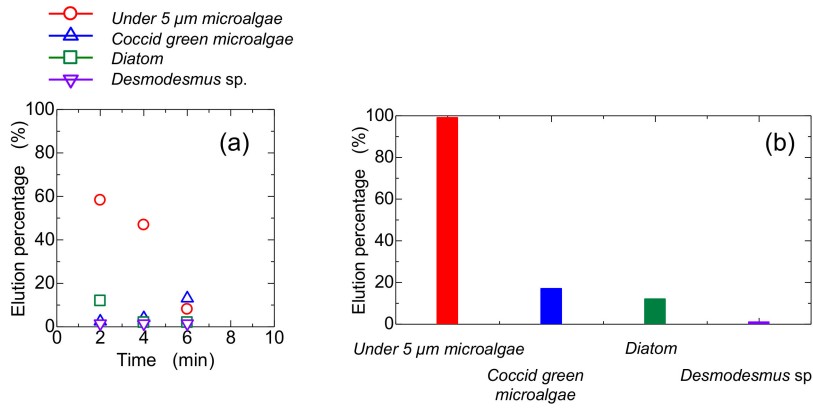

**Figure 8.** Conventional glass-packed column for elution of microalgae, (**a**) time course curves of each microalgae eluted, and (**b**) total elution percentage.

### 3.4. Dynamic Elution of Microalgae in Convex Change of Applied Magnetic Field

The bed of Mag gel was dynamically changed by applying a magnetic field in a convex manner to examine the elution of four microalgae. The convex magnetic field was applied at 10 min intervals. The elution behavior is shown in Figure 9. The *Coccid* green microalgae did not elute at all. The *Diatom* and *Desmodesmus* sp. eluted at 30 min. In the case of microalgae at a size of 5 μm and 0.4 T applied, the initial elution percentage increased then decreased. With an increase in the dynamic magnetic field, the smaller-sized microalgae eluted since the dynamic magnetic field induced the structure of the bed to be gradually tightened. The total elution percentage after 60 min is summarized in Figure 10, along with the constant magnetic field of 1.1 T. At a lower dynamic magnetic field of 1.1 T $\rightleftarrows$ 0.4 T and 1.1 T $\rightleftarrows$ 0.6 T, only the smaller-sized microalgae eluted. At 1.1 T $\rightleftarrows$ 0.8 T, the smaller-sized microalgae and *Diatom* and *Desmodesmus* sp. were eluted. Compared with constant 1.1 T, the elution percentage of the smaller-sized microalgae decreased, and the elution percentage of *Diatom* and *Desmodesmus* sp. was reversed.

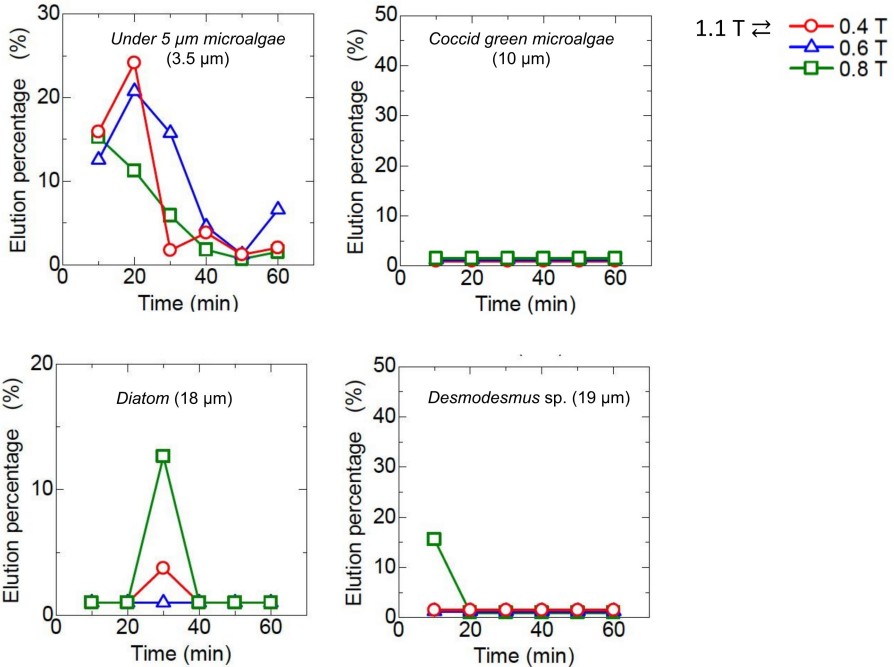

**Figure 9.** Dynamic elution of microalgae by a concave-like applied magnetic field. The magnetic field in the dynamic convex manner was changed at each interval of 10 min.

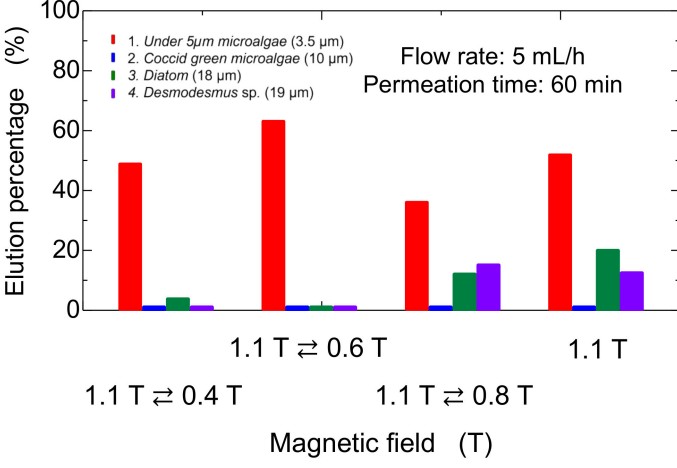

**Figure 10.** Dynamic elution of microalgae from Mag-gel-packed column at different magnetic fields.

A change of the bed of Mag gel by a dynamic magnetic field is illustrated in Figure 11. In Figure 11a, with about less than 5 μm of microalgae at 1.1 T, the path in the bed was formed to be eluted. Whereas at lower magnetic fields, the smaller-sized microalgae were gradually eluted through the gaps. In Figure 11b, in the *Diatom* and *Desmodesmus* sp., as the bed of Mag gel was compacted to the wall, the larger-sized microalgae were eluted, and the smaller microalgae gradually eluted at 1.1 T ⇌ 0.4 T, and 1.1 T ⇌ 0.6 T.

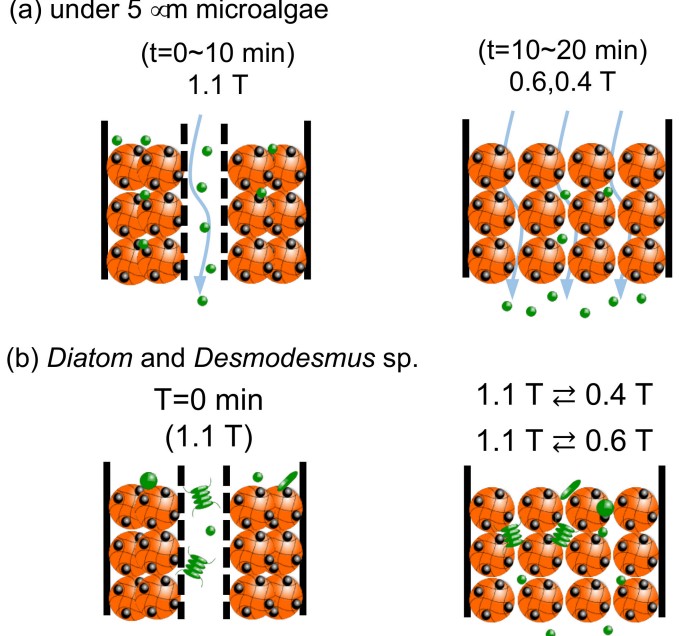

**Figure 11.** Illustrated image of dynamic elution of microalgae.(**a**) case of microalgae under the size of 5 micrometer, and (**b**) case of *Diatom* and *Desmodesmus* sp.

Flow cytometry is a popular technique for separating microalgae [24,25]. This separates microorganisms and cells by the shape and the color of analysis of observed images in the irradiation of lasers. Microalgae ability can be analyzed from the observed image with machine learning at present [26]. However, the applied laser would destroy the included colored proteins by the increments of heat. Some microalgae live in the sea, and the separation of several species is required. Different from the separation in water media, in terms of the separation of the microalgae that lived at sea, the effect of salt on the shrinkage of Mag gel should be considered, as where the bed was composed of Mag gel in salt solution changed the permeability of water. Thus, the gaps among the Mag gels as well as the flow of convective transfer of microalgae were altered.

The dynamic changes of the packed Mag gel stress the microalgae by shear and pressure in the narrowed pore. The observed images of the microalgae demonstrated that the deformation and disruption of the microalgae did not occur. Chromatographic analysis followed with the dynamic bed controlled by the applied magnetic field enabled us to observe the sophisticated separation of microalgae without disruption.

## 4. Conclusions

Magnetite-containing gel (Mag gel) was prepared by suspension polymerization at a size of 190 μm. The obtained Mag gel was packed in the column by applying a magnetic field to examine the separation of various microalgae via size and morphology. The applied magnetic field was set in a constant and convex manner to change the bed of Mag gel statically and dynamically, respectively. Microalgae, including four species, were cultured at Saga University. At a static magnetic field of 0.4–1.1 T, the smaller-sized microalgae eluted selectively. By increasing the applied constant magnetic field, the elution percentage of smaller-sized microalgae results from the change of bed. In a convex dynamic change

from 1.1 T, at the minor convex change of the magnetic field, the smaller-sized microalgae were eluted, whereas, in a more convex magnetic field, the larger-sized microalgae were selectively eluted. The dynamic change of magnetic field induced the change of bed structure composed of the Mag gel to enhance the elution selectivity of the microalgae, leading to the extraction of potential microalgae.

**Supplementary Materials:** The following supporting information can be downloaded at: https://www.mdpi.com/article/10.3390/separations9050120/s1. Figure S1: Images and size of obtained Mag gel, and the response of Mag gel to magnet, text: Supplementary Materials S2: Calculation of gel number in suspension and the amount of magnetite in gel. Figure S3: Packed structure of Mag gel in a column with the application of the magnetic field.

**Author Contributions:** Conceptualization, H.K. and M.D.; methodology, T.W.; software, T.W. and H.K.; validation, S.M., K.O. and H.K.; formal analysis, T.W.; investigation, T.W.; resources, M.D. and H.K.; data curation, H.K.; writing—original draft preparation, T.W.; writing—review and editing, H.K.; visualization, H.K.; supervision, H.K.; funding acquisition, H.K. All authors have read and agreed to the published version of the manuscript.

**Funding:** This research was funded by the Takahashi Industrial and Economic Research Foundation, Japan, and partially supported by funding from Saga city, Japan.

**Institutional Review Board Statement:** Not applicable.

**Informed Consent Statement:** Not applicable.

**Data Availability Statement:** Not applicable.

**Acknowledgments:** This study was partially supported by the Saga University Algae Research Project, Saga University, Japan, and the Analytical Research Center for Experimental Sciences, Saga University.

**Conflicts of Interest:** The authors declare no conflict of interest.

### Appendix A

Diagram of dynamic elution of microalgae by the application of magnetic field followed the below scheme.

(1) The mag gel was packed in the column; (2) the microalgae, including mainly four kinds of species, were injected on the top of the Mag gel layer; (3) water flowed through the Mag gel layer at various pressures; (4) the applied magnetic field was changed in a convex-like manner at an interval of 10 min; and (5) the elution was collected for the analysis of the microalgae.

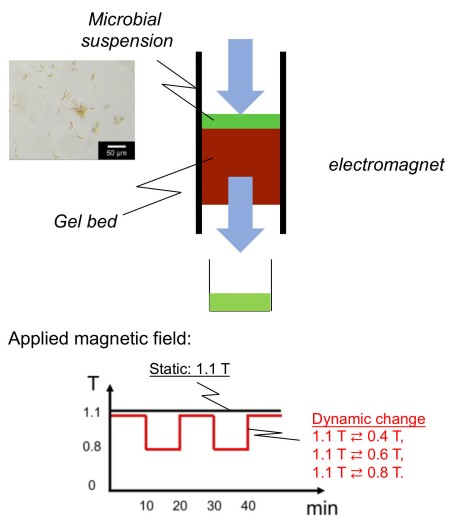

**Figure A1.** Illustrated equipment of bed of Mag gel in the dynamic application of a magnetic field.

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
