# Peer review of "Separation of Microalgae by a Dynamic Bed of Magnetite-Containing Gel in the Application of a Magnetic Field"

_separations, doi:10.3390/separations9050120_

Round 1
Reviewer 1 Report
- The introduction to the article is very limited in the literature on the application of the magnetic field, it is necessary to add more citations and the possibilities of using the magnetic field.
- The text is not justified in some places, this needs to be fixed.
- Figure 2 could be enlarged because it is not legible.
- Table 1 is not formatted according to the guidelines
- Figure 3 would be good to divide into 3 figures and refer to each of them in the text of the article.
- The formulas in lines 131 and 149 must be properly formatted and quoted in the text of the article.
- Figures 4 and 5 should be moved to the place where they are described.
- In the materials and methods section, you need to add a diagram of the experiment in the form of a drawing with captions.
- The conclusions should justify the text and describe the possibilities of using the magnetic field to separate microalgae, which may bring benefits in the future, or the possibilities of developing this application.
Author Response
Reviewer Comments 1
1 The introduction to the article is very limited in the literature on the application of the magnetic field, it is necessary to add more citations and the possibilities of using the magnetic field.
Reply: Thanks to the reviewer comments, the following sentences regarding the magnetite application for the recovery of microalgae industry are inserted.
Magnetite particles thus far have been used for harvesting microalgae, and purification of protein, polysaccharides, and cells. Functional group was introduced to the surface of magnetite particle. The obtained particle was added to the solution and after the adsorption, magnetic particle was recovered by the application of magnetic field. This technique was mainly used in batch mode.
Liu, Y.; Jin, W.; Zhou, X.; Han, S.-F.; Tu, R.; Feng, X.; Jensen, P. D.; Wang, Q. Efficient harvesting of Chlorella pyrenoidosa and Scenedesmus obliquus cultivated in urban sewage by magnetic flocculation using nano-Fe3O4 coated with polyethyleneimide. Biores. Technol. 2019, 290, 121771.
Chen, Y.; Jiang, P.; Liu, S.; Zhao, H.; Cui, Y.; Qin, S. Purification of 6×His-tagged phycobiliprotein using zinc-decorated silica-coated magnetic nanoparticles. J. Chromatogr. B 2011, 879, 993.
Franzreb, M.; Siemann-Herzberg, M.; Hobley, T.J.; Thomas, O. R. T. Protein purification using magnetic adsorbent particles. Appl. Microbiol. Biotechnol. 2006, 70, 505.
Mohapatra, S.; Panda, N.; Pramanik, P. Boronic acid functionalized superparamagnetic iron oxide nanoparticle as a novel tool for adsorption of sugar. Mater. Sci. Eng. 2009, C 29, 2254.
Raghavarao, K. S. M. S.; Dueser, M.; Todd, P. Multistage magnetic and electrophoretic extraction of cells, particles and macromolecules. Advanced Biochem. Eng./Biotechnol. 2000, 68, 139.
2 The text is not justified in some places, this needs to be fixed.
Reply: According to the reviewer suggestion, some sentences are revised. Thank you very much.
3 Figure 2 could be enlarged because it is not legible.
Reply: Thanks to the reviewer comments, Figure 2 is enlarged to consider the readability.
4 Table 1are not formatted according to the guidelines
Reply: As the reviewer suggestion, Tables 1 and 2 are revised.
5 Figure 3 would be good to divide into 3 figures and refer to each of them in the text of the article.
Reply: The packed structure of Mag gel in column is critical for separation of microalgae. Packed structure was strongly dependent of the flow rate of water, magnitude of magnetic field, and number of Mag gel in column. The authors want to demonstrate these effects in one figure, thus figures in Figure 3 are not divided. We are appreciated to the suggestion of the reviewer.
6 The formulas in lines 131 and 149 must be properly formatted and quoted in the text of the article.
Reply: Thanks to the reviewer comments, format of definition of eqs. 1) and 2) are revised and the number of eqs. 1) and 2) are inserted to the text.
amount of Mag gel were changed to examine the packing stability of bed from the following leakage percentage, as eq. 1), of Mag gel.
Elution percentage of each microalgae was determined by the following equation 2),
7 Figures 4 and 5 should be moved to the place where they are described.
Reply: Thanks to the reviewer suggestion, figures are moved to the appropriate position for readerability.
8 In the materials and methods section, you need to add a diagram of the experiment in the form of a drawing with captions.
Reply: Thanks to the reviewer’s suggestion, the experimental flow for dynamic separation of microalgae with bed of Mag gel are added to Appendix A.
Appendix A
Diagram of dynamic elution of microalgae by application of magnetic field followed the below scheme.
1) Mag gel was packed in column, 2) microalgae including mainly four kinds of species was injected on the top of Mag gel layer, 3) water was flowed through Mag gel layer at various pressures, 4) applied magnetic field was changed with convex -like manner at the interval of 10 min, and 5) elution was collected for analysis of microalgae.
Figure A-1. Illustrated equipment of bed of Mag gel in dynamic application of magnetic field.
9 The conclusions should justify the text and describe the possibilities of using the magnetic field to separate microalgae, which may bring benefits in the future, or the possibilities of developing this application.
Reply: Thanks to the reviewer comments, the potential application of dynamic bed composed of Mag gel, as below.
Homogeneous structure of the bed in column has been used for the purification of microalgae, as well as biomolecules such as proteins, and polysaccharides. In this study, dynamic change of magnetic field induces the change of bed structure composed of Mag gel, to enhance the elution selectivity of microalgae based on their size and interaction with gel. This dynamics of the bed also are applicable for purification of protein and exosome to change the interaction with Mag gel.
Reviewer 2 Report
The paper deals with the studies on separation of microalgae organisms by employing magnetite-containing gel under the conditions of external magnetic field. The elution efficiency was examined in function of the magnitude of the applied magnetic field. The paper presents some information of interest to the readers but several issues require deeper clarification.
- Adsorption of microalgae on Mag gel – more quantitative information is required (adsorption isotherm, capacity, kinetics).
- Real samples of microalgae living e.g. in sea water are composed of quite complicated chemical matrix. The salinity of real samples may affect the structure of the polymeric gel (it may induce shrinkage of the gel). Was the separation procedure validated for real samples? More information is required.
- Visualization of the real experimental setup of the magnetite-containing gel packed in column by the application of magnetic field would be very helpful for potential readers.
Author Response
Reviewer comments 2
The paper deals with the studies on separation of microalgae organisms by employing magnetite-containing gel under the conditions of external magnetic field. The elution efficiency was examined in function of the magnitude of the applied magnetic field. The paper presents some information of interest to the readers but several issues require deeper clarification.
1 Adsorption of microalgae on Mag gel – more quantitative information is required (adsorption isotherm, capacity, kinetics).
Reply: As the reviewer’s comments, isotherm curve of Mag gel to microalgae is shown in Figure 4. The ratio of microalgae to one Mag gel was mentioned in the text.
2 Real samples of microalgae living e.g. in sea water are composed of quite complicated chemical matrix. The salinity of real samples may affect the structure of the polymeric gel (it may induce shrinkage of the gel). Was the separation procedure validated for real samples? More information is required.
Reply: Thanks to the suggestion of the reviewer, the following sentences are inserted to the text for separation of microalgae lived in the sea.
Some microalgae lived in the sea, and separation of several species is required. Different from the separation in water media, in the separation of the microalgae lived in the sea should be considered the effect of salt to shrinkage of Mag gel, where the bed composed of Mag gel in solution of salt changed the permeability of water, thus gaps among Mag gels as well as flow of convective transfer of microalgae are altered.
3 Visualization of the real experimental setup of the magnetite-containing gel packed in column by the application of magnetic field would be very helpful for potential readers.
Reply: Deformation of Mag gel in application of Magnetic field was not yet observed directly. Experimental equipment is mentioned in Supplementary materials 3.
Round 2
Reviewer 1 Report
Everything has been corrected in line with the comments. I recommend the article for publication.
Author Response
Thank you very much.
Reviewer 2 Report
I cannot find the sentences (below) in the text...
"Some microalgae lived in the sea, and separation of several species is required. Different from the separation in water media, in the separation of the microalgae lived in the sea should be considered the effect of salt to shrinkage of Mag gel, where the bed composed of Mag gel in solution of salt changed the permeability of water, thus gaps among Mag gels as well as flow of convective transfer of microalgae are altered."
Author Response
Thanks to the reviewer's help, the following sentences are inserted. Sorry for forgetting the insert.
Some microalgae lived in the sea, and separation of several species is required. Different from the separation in water media, in the separation of the microalgae lived in the sea should be considered the effect of salt to shrinkage of Mag gel, where the bed composed of Mag gel in solution of salt changed the permeability of water, thus gaps among Mag gels as well as flow of convective transfer of microalgae are altered